# Antibody–Drug Conjugates for the Treatment of Acute Pediatric Leukemia

**DOI:** 10.3390/jcm10163556

**Published:** 2021-08-13

**Authors:** Jamie L. Stokke, Deepa Bhojwani

**Affiliations:** 1Division of Hematology-Oncology, Children’s Hospital Los Angeles, Keck School of Medicine, University of Southern California, Los Angeles, CA 90027, USA; dbhojwani@chla.usc.edu; 2Norris Comprehensive Cancer Center, University of Southern California, Los Angeles, CA 90033, USA

**Keywords:** ADC, antibody–drug conjugate, pediatric leukemia, leukemia, ALL, AML, immunotherapy

## Abstract

The clinical development of antibody–drug conjugates (ADCs) has gained momentum in recent years and these agents are gradually moving into frontline regimens for pediatric acute leukemias. ADCs consist of a monoclonal antibody attached to a cytotoxic payload by a cleavable linker. This structure allows for highly cytotoxic agents to be directly delivered to leukemia cells leading to cell death and avoids excessive off-tumor toxicity. Near universal expression on B-cell acute lymphoblastic leukemia (ALL) blasts and the ability of rapid internalization has rendered CD22 an ideal target for ADC in B-ALL. Inotuzumab ozogamicin, the anti-CD22 antibody linked to calicheamicin led to complete remission rates of 60–80% in patients with relapsed/refractory B-ALL. In acute myeloid leukemia (AML), the CD33 targeting gemtuzumab ozogamicin has demonstrated modest improvements in survival and is the only ADC currently licensed in the United States for pediatric patients with de novo AML. Several other ADCs have been developed and tested clinically for leukemia but have achieved limited success to date. The search for additional leukemia-specific targets and optimization of ADC structure and specificity are ongoing efforts to improve their therapeutic window. This review provides a comprehensive overview of ADCs in acute leukemias, with a focus on pediatric ALL and AML.

## 1. Introduction

The outcome of pediatric patients with leukemia has improved dramatically in recent decades with overall survival exceeding 90% in B-cell acute lymphoblastic leukemia (B-ALL) [1]. Modest improvements have been noted in acute myeloid leukemia (AML) with overall survival of greater than 60% [2]. However, survival for patients with high-risk and relapsed leukemias is much lower, and despite highly toxic, intensified therapies, durable remission is difficult to achieve. In addition, patients suffer significant toxicities related to intensive chemotherapy regimens. Thus, targeted agents are crucial, and several have been developed in the relapsed setting. A few of these agents are now being tested in frontline therapy, with goals of improving outcomes and mitigating short and long-term toxicity. Targeted immunotherapy utilizing antibodies, antibody–drug conjugates (ADCs), immunotoxins, bi-specific antibody T cell engagers (BiTEs), and chimeric antigen receptor (CAR) T cells have changed the treatment landscape for relapsed and high-risk B-ALL. Similar success with targeted therapies has been slower in AML due to disease heterogeneity and the potential for high off-tumor toxicity associated with target antigen expression in normal hematopoietic stem cells resulting in myeloablation.

ADCs are a promising and rapidly expanding repertoire of oncology therapeutics. They offer an effective mechanism of delivering highly cytotoxic agents directly to leukemia cells and avoiding off-target toxicity seen with standard chemotherapy. ADCs consist of a monoclonal antibody bound to chemotherapeutic drugs (payload) via chemical linkers. Upon antibody binding to its target on a leukemia cell, the ADC is internalized via receptor-mediated endocytosis. The linker is then cleaved, and the cytotoxic drugs are released inside the cell. Depending on the drug delivered, there are multiple mechanisms of action leading to apoptosis and cell death (Figure 1). The cytotoxic agent of the ADCs also has the ability to be cleaved in the tumor microenvironment and cross cell membranes of neighboring tumor cells, known as the “bystander effect”.

Multiple generations of ADCs have evolved with improved stability, potency, and internalization kinetics [3,4]. Gemtuzumab ozogamicin (GO) is a first-generation ADC targeting CD33 on myeloid leukemia cells and the first to attain United States Food and Drug Administration (FDA) approval for adult patients with relapsed AML in 2000. Though subsequent development was not straightforward, the indication was extended to newly diagnosed adult patients in 2017, and to newly diagnosed pediatric patients in 2020. At the time of this review, there are six ADCs approved by the US FDA for hematologic malignancies: gemtuzumab ozogamicin (CD33), brentuximab vedotin (CD30), inotuzumab ozogamicin (CD22), polatuzumab vedotin (CD79B), belantamab mafodotin (BCMA) and loncastuximab tesirine (CD19). The first five have also been approved by the European Medicines Agency (EMA); loncastuximab tesirine is awaiting approval [4,5]. This review highlights ADCs developed for clinical use in acute leukemias. Table 1 details ongoing clinical trials.

### Antibody–Drug Conjugate Design

ADC development requires attention to stability in physiologic conditions and careful consideration for the choice of the target antigen. The target should be sufficiently expressed on the leukemia cell surface, only expressed in low levels on healthy tissues, and rapidly internalized upon antibody binding [6]. The antibody that binds to the antigen must have a suitable affinity to allow for sufficient binding and internalization. Most antibodies used clinically are human IgG to limit immunoreactivity. The linker attaching the antibody to the payload must be stable during circulation. Upon endocytosis, the linker is broken down within the cell by either enzymatic reaction or hydrolyzed by pH conditions, and the cytotoxic drug is released [4]. The cytotoxic agents must also be stable to avoid degeneration prior to reaching their targets. There are two main categories of cytotoxic drugs used in ADCs as payloads: microtubule inhibitors and DNA-damaging drugs. DNA damage can occur via double-stranded DNA breakage (calicheamicin), by alkylating DNA (duocarmycin), or by crosslinking with DNA (pyrrolobenzodiazepine dimers). An example of ADCs that utilize microtubule inhibitors is brentuximab vedotin, and those which induce DNA damage include InO and GO. These cytotoxic agents demonstrate much higher potency than traditional chemotherapeutic agents, and thus, targeted and stable delivery is the key to their clinical success [7].

Despite the benefit of a targeted approach to limit the toxicity of ADCs, there are multiple mechanisms of on and off-target tissue damage. As target antigens may also be expressed in certain healthy tissues, ADCs can result in toxicities from on-target, off-tumor targeting. Also, linker instability can lead to the premature release of the cytotoxic payload into the circulation. Toxicities include those commonly seen with standard chemotherapeutic agents and include anemia, neutropenia, thrombocytopenia, hepatic toxicity, and peripheral neuropathy [8]. For example, both InO and GO are known to cause hepatotoxicity including elevated transaminases, hyperbilirubinemia, and sinusoidal obstructive syndrome (SOS) attributed to calicheamicin. The mechanisms of hepatotoxicity for these agents are multifactorial including on-target, off-tumor antibody binding, nonspecific uptake of the ADCs in the liver, and premature release of the payload into circulation [9]. New generation ADCs are in development with the goals of improved safety and efficacy. For instance, newer antibodies are typically IgG1 subclass to optimize their solubility, target affinity, and half-life. Target selection has improved over time, with the focus placed on higher rates of turnover for increased antitumor activity, and targets that are oncogenic are also being explored. The linker moiety has evolved with the improvement in stability in circulation to limit off-target toxicities. Lastly, the payload itself has undergone engineering improvements. Having a high payload to antibody ratio and utilizing hydrophilic constructs leads to great antitumor activity and avoidance of hepatic clearance [10].

## 2. Antibody–Drug Conjugates in B-ALL

### 2.1. Targeting CD22

CD22 is a regulator of B cell signaling and is expressed on 96% of B-ALL blasts [11]. Its B cell-specific expression makes it an ideal target for antibody therapy [12]. In addition, the CD22 receptor is internalized rapidly. Inotuzumab ozogamicin (InO) was approved by the FDA and EMA in 2017 to treat adult patients with relapsed or refractory B-ALL [13]. The structure of InO consists of the antitumor antibiotic calicheamicin, a cleavable linker, and an ant-CD22 IgG4 antibody. Once in the nucleus, calicheamicin causes double-stranded DNA breaks leading to apoptosis and cell death. Additionally, calicheamicin diffuses outside of the cell and into neighboring cancer cells leading to cytotoxicity via the bystander effect [4].

InO has demonstrated excellent efficacy in adults with B-ALL. In the phase III INO-VATE trial comparing single-agent InO to standard chemotherapy for patients with relapsed B-ALL, the rate of complete remission (CR) was significantly higher in the InO group than the chemotherapy group (80% vs. 29%) [14]. In pediatric patients, outcomes have been equally promising. In a compassionate use program, 51 children received InO and 67% of those responders achieve minimal residual disease (MRD) negativity [15]. In the COG study, AALL1621 (NCT02981628), InO was administered in a single-arm phase II trial for patients with multiply relapsed or refractory B-ALL. A CR rate of 58% was achieved [16]. The Innovative Therapies for Children with Cancer (ITCC) Consortium, recently completed a phase I study in multiply relapsed patients with B-ALL. Overall response rate (ORR) after course 1 was 80% and 84% of responders achieved MRD negative remission [17]. To improve on the durability of response and test safety in combination with standard chemotherapy, the ongoing ITCC consortium and the COG trials combine InO with cytotoxic chemotherapy for patients with relapsed/refractory ALL. Additionally, the ongoing COG trial AALL1732 (NCT03959085) for newly diagnosed patients is testing single-agent InO courses between standard chemotherapy phases versus chemotherapy alone. Important adverse events (AEs) of InO include SOS which occurred in 11% of patients in the INO-VATE trial with most cases occurring after hematopoietic stem cell transplant (HSCT) [13,18]. The post-HSCT SOS rate was higher in pediatric patients; a plausible reason is the inclusion of very heavily pretreated patients in the pediatric cohorts [15,16]. Prophylactic therapies such as ursodiol and defibrotide may be warranted for patients who will undergo post-InO HSCT. Other serious AEs of InO include neutropenia, thrombocytopenia, febrile neutropenia, infusion-related reactions, tumor lysis syndrome, and prolonged QT syndrome [13,18]. Despite the high CR rate of InO, the response is suboptimal in those with baseline dim or partial CD22 expression and in patients with *KMT2A* rearrangements [19]. In addition, modulation of CD22 expression and emergence of CD22 negative clones is a mechanism of resistance post InO.

Another ADC targeting CD22, Epratuzumab tesirine, or ADCT 602, composed of an anti-CD22 humanized IgG1 antibody bound to a pyrrolobenzodiazepine (PBD) dimer (a DNA crosslinking agent) via a cleavable linker is currently under investigation in a phase I/II clinical trial in adults with relapsed and refractory B-ALL (NCT03698552) [20].

Immunotoxins are antibody-protein toxin conjugates which, similar to ADCs, utilize the specific binding power of antibodies to deliver toxins derived from bacteria, fungi, and plants. They function by inhibiting protein synthesis, and an antibody fragment is used rather than an entire antibody to allow for improved pharmacokinetics [21]. HA22, Moxetumomab pasudotox is a recombinant CD22 targeting immunotoxin utilizing a pseudomonas exotoxin (PE) [22]. In a phase I trial, approximately 23% of children with relapsed ALL achieved a complete response and toxicities included capillary leak syndrome (CLS) and HUS [23]. In an international Phase 2 study, 28 of 32 enrolled patients were evaluated for a response, and the ORR was 28% with 10% of patients achieving a morphologic CR. However, the study was terminated early as the CR rate was suboptimal and did not achieve the target [24]. In a parallel phase 2 trial administering Moxetumomab to pediatric ALL patients with positive MRD prior to HSCT (12-MOXE), the sole patient enrolled developed fatal CLS and this study was also terminated [25]. Moxetumomab has an acceptable safety profile in adult patients with relapsed/refractory hairy cell leukemia (HCL) and is FDA approved for this indication [26]. Though HUS and CLS are noted in patients with HCL, these toxicities are transient in most patients. It is unclear if the mechanisms of toxicity differ based on the host (pediatric vs. adult patients) or the disease (ALL vs. HCL).

### 2.2. Targeting CD19

CD19 is expressed on normal and neoplastic B cells and plays a key role in B-cell signaling, activation, and B-cell development. CD19 is ubiquitously expressed on B-ALL cells and has served as an effective target for antibody therapy. However, it is not internalized as rapidly as CD22, thus, is less efficient in drug delivery despite its high cell surface density. Loncastuximab tesirine, or ADCT-402, is an ADC consisting of a humanized anti-CD19 antibody conjugated to SG3199, a PBD dimer-containing toxin [20]. Loncastuximab has demonstrated safety and efficacy in CD19 positive non-Hodgkin lymphomas resulting in an ORR of 45% in the phase I study, and 48% in the phase II study [5,27]. However, in the phase I study of loncasutximab for adults with relapsed or refractory B-ALL, only 3 of 35 patients (8.6%) achieve CR, and the trial closed early due to slow accrual [28]. Common toxicities included nausea, febrile neutropenia, and liver abnormalities [27].

Denintuxumab mafodotin, or SGN-CD19A, is a humanized anti-CD19 monoclonal antibody conjugated to monomethyl auristatin F (MMAF). In vitro studies demonstrated activity against pediatric ALL by delaying progression in eight patient-derived xenografts [29]. A phase I dose-escalation study to assess safety in adult patients with relapsed or refractory B-ALL ended in 2017 and an interim report of this trial noted a 19% CR rate [30]. Phase II studies in combination with chemotherapy for lymphoma were initiated but terminated early due to changes in portfolio prioritization by the sponsor.

Coltuximab ravtansine, or SAR3419, is a humanized CD19 antibody with a maytanisoid DM4 payload. Preclinical studies examined SAR3419 alone or in combination with chemotherapy in pediatric patient-derived xenografts and demonstrated an objective response in all but one xenograft prompting its development into a clinical trial [31]. The MYRALL trial was a phase II monotherapy study in 36 adults with relapsed or refractory ALL. At the recommended dose of 70 mg/m^2^, 3 of 17 patients attained CR with a duration of response of 1.9 months. The most common toxicities were fever, diarrhea, and nausea. Due to an inadequate response rate, this study was terminated [32].

### 2.3. Targeting CD25

CD25 is expressed on activated B cells, T cells, and regulatory T cells and is the alpha chain of the IL-2 receptor. Expression on ALL and AML is associated with induction failure, increased risk of relapse, and decreased overall survival [20]. Camidanlumab tesirine, or ADCT-301, is a humanized IgG1 anti-CD25 antibody conjugated to a PBD dimer. In a phase I trial of adult patients with classical Hodgkin lymphoma, the ORR rate was impressive at 81% [33]. Unfortunately, the phase I trial of adults with CD25 positive relapsed or refractory ALL or AML was terminated early due to limited efficacy as only 2 of 35 patients achieved CR. Common toxicities included febrile neutropenia, cytopenias, fatigue, pneumonia, hypophosphatemia, and elevated gamma glutamyltransferase [34,35]. This agent is continuing development as a mechanism to deplete regulatory T cells as a single agent, and in combination with checkpoint inhibitors in solid tumors [36].

### 2.4. Current Clinical Applications of ADCs in Pediatric B-ALL

The therapeutic approach for relapsed B-ALL varies but most pediatric patients with first relapse B-ALL are treated with a standard four-drug re-induction. Therapy following re-induction is dependent on risk stratification but typically consists of either blinatumomab and chemotherapy for lower-risk patients or chemotherapy followed by HSCT for higher-risk patients [37]. InO has demonstrated efficacy in second or greater relapse (or refractory disease) in both COG trial AALL1621 and the European ITCC trial and is a good option as a bridge to transplant in that patient population [16,17]. InO is also a useful agent as a re-induction regimen for patients with CD19 antigen-negative relapse B-ALL who do not otherwise qualify for CD19 directed CAR T cells or blinatumomab. CD22 targeting CAR T cells and InO demonstrate similar efficacy in relapsed and refractory B-ALL, however, InO is easier to administer as it is available off the shelf and given as a once-weekly IV infusion [38]. CD22 CAR T cells, on the other hand, are still in the early phases of development and are only available in the context of a clinical trial at few centers requiring weeks for manufacturing. Special consideration should be taken when considering salvage InO prior to HSCT due to the risk of SOS which is highest in heavily pre-treated patients. InO is currently being studied as a frontline agent in trials for high-risk B-ALL which combine InO with the standard BFM chemotherapy backbone. Other ADCs in development are currently limited to phase I/II trials in adult patients.

## 3. Role of Antibody–Drug Conjugates in T-ALL

Development of successful immunotherapy for T-ALL is challenging due to the shared expression of target antigens between normal and leukemic T cells, and toxicities associated with T cell depletion. CAR T cells are currently under development targeting several T cell antigens including CD5, CD7, CD3, and CD4, but ADCs have lagged behind [39]. CD30 expression is noted in 38% of T-ALL cases, with increased expression observed during courses of chemotherapy [40]. Brentuximab vedotin is approved for cutaneous T cell lymphoma, but no studies have been initiated in T-ALL yet. Monoclonal antibodies targeting T-ALL antigens have also shown clinical promise. Daratumumab, an anti-CD38 monoclonal antibody, has demonstrated efficacy in some patients with T-ALL [41]. Other potential targets for ADCs include IL7R which is a transmembrane receptor that plays a role in the maintenance and progression of T-ALL. Preclinical models demonstrated increased steroid sensitivity in lymphoid blasts by targeting IL7R with the ADC A7R-ADC-SN-38, and a clinical trial is in the early stages of development [42].

## 4. Antibody–Drug Conjugates in AML

### 4.1. Targeting CD33

CD33 is variably expressed on the majority of leukemic myeloblasts, and high CD33 expression is associated with an inferior outcome [43]. Gemtuzumab ozogamicin (GO) is the first approved ADC for human use [44]. GO is an anti-D33 IgG4 antibody linked to the calicheamicin cytotoxic agent. In the first phase I clinical trial of GO administered to 40 adults with relapsed/refractory AML, a CR rate of 12.5% was achieved [45]. In a follow-up report consisting of three single-arm phase II studies, 270 adult patients in the first relapse were enrolled and 71 (26%) achieved CR with single-agent GO [46]. These results led to the accelerated approval by the FDA in 2000 as a stand-alone treatment of patients over 60 years of age who were not candidates for standard chemotherapy [46]. GO was later withdrawn from the commercial market in October 2010 after a randomized trial examining standard chemotherapy versus chemotherapy with GO in 637 adult patients showed no improvement in survival and increased treatment-related toxicity in the GO arm [47]. However, after additional data including the pivotal ALFA-0701 trial by the Acute Leukemia French Association, the FDA reapproved the use of GO in 2017 [48].

In pediatric patients, the Berlin-Frankfurt-Münster (BFM) group first demonstrated a 4-year overall survival of 18% in a GO compassionate use program followed by a phase II study of GO resulting in a 37% CR/CRi in those with relapsed and refractory AML [49,50]. Subsequently, the UK Medical Research Council (MRC) AML15 trial demonstrated the efficacy of GO in upfront treatment of pediatric AML [51,52]. COG AAML03P1 added GO to standard chemotherapy followed by HSCT if the patient had a matched donor. The CR rate was 83% after 1 course and 87% after 2 courses [52]. The subsequent trial COG AAML0531 compared standard chemotherapy to standard chemotherapy with GO. In this trial, GO significantly improved event-free survival (53% vs. 46% in non-GO arm) but not overall survival [53]. These results supported the FDA approval of GO for pediatric patients aged 1 month and older. Data from the UK MRC15 and ALFA-0701 trials suggest that GO particularly benefits patients with favorable and intermediate-risk cytogenetics [51,54]. Toxicities in the pediatric trials include hyperbilirubinemia, elevated transaminases, SOS, febrile neutropenia, and prolonged neutrophil recovery [52,53,55]. The dosing of GO has evolved over time with infusion-related toxicities and SOS observed at 9 mg/m^2^ [56]. Another study demonstrated equivalent CR and less toxicity (including SOS) with a dosing of 3 mg/m^2^ compared to 6 mg/m^2^ [57]. In combination with chemotherapy, a single dose of 3 mg/m^2^ is commonly used in pediatric and adult practice. Several ongoing studies are examining the use of GO with different chemotherapy combinations. MyeChild01 is an ongoing European consortium trial evaluating the optimum tolerated number of 3 mg/m^2^ doses of GO to be used in combination with induction chemotherapy in pediatric patients (NCT02724163). AAML1831 (NCT04293562) is the ongoing phase III COG study comparing standard chemotherapy to therapy with CPX-351 in newly diagnosed children with AML. All patients receive GO in the backbone regimen as the standard of care.

To improve upon the efficacy and reduce toxicities of GO, additional ADCs targeting CD33 have been developed, but these continue to face multiple challenges. AVE9633 is an ADC composed of a highly potent maytansinoid derivative, DM4, conjugated to a humanized IgG1 anti-CD33 monoclonal antibody, huMy9-6. In a phase I trial in 54 adults with refractory or relapsed AML, the most common adverse event was an infusion-related reaction. Unfortunately, only two patients had a response; one CR for 8 months and another PR for two months [58]. Vadastuximab talirine, or SGN-CD33A, is an ADC consisting of a PBD dimer linked to an antibody targeting CD33. In a phase I study of 131 adult patients with CD33-positive AML, vadastuximab led to a 28% CR rate [59]. The subsequent phase III CASCADE study assessed vadastuximab in combination with hypomethylating agents (HMAs) compared to HMAs alone. A safety analysis indicated a higher rate of deaths, including fatal infections in the vadastuximab arm compared to the control arm, and the study was closed (NCT02785900). Continued development will incorporate additional safeguards and toxicity monitoring rules. IMGN779 is an anti-CD33 ADC with a DNA-alkylating IGN (indolinobenzodiazepine pseudodimer) payload and a cleavable s-SPDB linker. A phase I trial of IMGN779 enrolled 50 adult patients with relapsed or refractory AML, and the most common toxicities included febrile neutropenia, nausea, diarrhea, and fatigue. Overall, 41% of patients demonstrated a decrease in bone marrow blasts, but it is unclear if the development of this agent will proceed [60].

### 4.2. Targeting CD123

CD123 is the alpha subunit of the interleukin-3 receptor and is highly expressed in AML, blastic plasmacytoid dendritic cell neoplasm, B-ALL, and early thymic progenitor ALL cases [61,62]. It is rapidly internalized making it an ideal target for antibody therapy [63]. IMGN632 is an ADC consisting of a novel DNA alkylating payload, DGN549 which is an indolinobenzodiazepine pseudodimer (IGN) class, that induces single stranded DNA breaks and a novel peptide linker that confers greater stability in circulation [64]. A phase I study for adult patients with relapsed or refractory CD123 positive leukemia is actively recruiting to assess the safety and tolerability of IMGN632 as monotherapy. Preliminary analysis demonstrated a 33% CR rate, and common toxicities included diarrhea, nausea, febrile neutropenia, peripheral edema, and hypotension (NCT03386513) [65]. A phase Ib/II study for adult patients with CD123-positive AML utilizing IMGN632 as either monotherapy or in combination with venetoclax and/or azacytidine is actively recruiting (NCT04086264).

SGN-CD123A is an anti-CD123 antibody bound to a PBD dimer that was evaluated in a phase I study (NCT02848248) in adult patients with relapsed or refractory AML, however, this study was terminated early at the same time as the CD33 ADC vadastuximab talirine study for safety concerns as it utilized the identical PBD dimer and linker molecules that resulted in excessive toxicities.

### 4.3. Targeting ROR1

Receptor tyrosine kinase-like orphan receptor 1 (ROR1) is expressed on hematologic malignant cells but not on normal tissues. It is expressed in 35% of AML and in most cases, it is co-expressed with CD34, indicating it is a promising target for leukemia stem cells. The ADC VLS-101 consists of a humanized IgG1 monoclonal antibody (UC-961), and an mc-VC-PAB linker bound to the cytotoxic payload, MMAE [66,67]. An ongoing phase I trial examines the use of VLS-101 in adult patients with relapsed hematologic malignancies including ALL and AML (NCT03833180).

### 4.4. Targeting Mesothelin

Mesothelin (MSLN) is highly overexpressed in about 33% of pediatric AML cases and not in normal bone marrow making it a viable target [68]. Anetumab ravtensine, or BAY 94–9343, is an ADC consisting of anti-MSLN linked to tubulin polymerase inhibitor DM4. In view of a favorable safety profile for this agent in an adult trial of patients with advanced solid tumors, a COG phase I study is in development for second or greater relapse pediatric patients with mesothelin-positive AML [69,70].

### 4.5. Targeting CLL-1 (CD371)

C-type lectin-like molecule-1 (CLL-1) is a transmembrane glycoprotein expressed on the surface of AML blasts, AML stem cells, and monocytes but not on hematopoietic stem cells [71]. DCLL9718S is a THIOMAB^TM^ antibody-drug conjugated (TDC) consisting of an IgG1 anti-CLL1 antibody linked to two PBD dimers via a cleavable disulfide linker. THIOMAB^TM^ consists of engineering a recombinant mutation of one or more amino acids to a cysteine which allows the ADC to achieve improved stability of the connection of cytotoxic drug to antibody [72]. A clinical trial examined 18 adult patients with relapsed or refractory AML in a phase I trial of DCLL9718S [73]. Two-thirds of the patients experienced at least one clinically significant AE most commonly including febrile neutropenia and pneumonia. No patients achieved objective CR or PR response. Due to the limited tolerability and efficacy, this drug will not move forward in clinical trials, however, CLL-1 remains a promising target for CAR T-cell therapy and future ADCs [74].

### 4.6. Additional ADCs with Unclear Clinical Potential in AML

CD30 is a cell membrane protein of the tumor necrosis factor receptor family expressed on activated T and B cells, and also on 36% of high-risk AML/MDS [75]. Brentuximab vedotin is an ADC consisting of an anti-CD30 antibody conjugated to the anti-microtubule compound, MMAE. After licensing in Hodgkin lymphoma and anaplastic large cell lymphoma, brentuximab was studied in other CD30 expressing hematologic malignancies. In a phase I study in adults with AML, brentuximab was combined with re-induction chemotherapy in CD30 expressing relapsed AML [76]. The composite response rate was 36% with a median disease-free survival of 6.8 months [76]. A phase I/II study examining brentuximab with azacytidine in AML was terminated early due to poor accrual (NCT02096042). A phase II study of single-agent brentuximab in CD30 positive non-lymphomatous malignancies enrolled pediatric and adult patients with AML and solid tumors (NCT01461538). According to the preliminary report, 2 of 14 patients with leukemia or high-grade myelodysplastic syndrome (MDS) achieved the objective response [77]. Final results are awaited. CD37 is another transmembrane protein that is highly expressed on myeloid cells and may function as a signaling death receptor. The ADC AGS67E is a humanized monoclonal IgG3 antibody against CD37 conjugated via a protease-cleavable linker to MMAE [78]. In a phase I study, 23 adult patients with AML were enrolled, however, the study was terminated due to business reasons (NCT02610062).

Members of the receptor tyrosine kinase family c-KIT and FLT3 have been amenable to targeting with small molecule inhibitors, and CAR T-cells against these targets are also being developed. Unfortunately, the clinical development of ADCs for c-KIT (CD117) and FLT3 (CD135) has been difficult. LOP628 is a humanized antibody against CD117, conjugated to DM1 via a non-cleavable linker [79]. A phase I study of LOP628 in patients with c-KIT positive solid tumors and AML enrolled three participants and then closed early due to infusion reaction in the first two patients due to mast cell degranulation [80]. AGS62P1 is an ADC consisting of an anti-FLT3 human IgG1 antibody conjugated to a microtubule-disrupting agent (AGL-0182-30) via an alkoxyamine linker [81]. A phase I study enrolled 43 participants with relapsed and/or refractory AML was terminated early due to lack of efficacy (NCT02864290).

### 4.7. Current Clinical Applications of ADCs in Pediatric AML

GO has demonstrated the highest clinical success among ADCs in AML and is cur-rently used in combination with standard chemotherapy in frontline studies. GO is FDA approved for the treatment of newly diagnosed CD33 positive AML in children ≥2 years of age and thus is often utilized with many different standard chemotherapy regimens in pediatric patients with CD33 positive AML. In the relapsed setting, GO is often used in combination with standard chemotherapy as in the frontline setting. In patients with both frontline and relapse use of GO, careful consideration must be used when administering GO prior to HSCT to decrease the risk of SOS. The risk of SOS is decreased when a lower dose (3 mg/m^2^) is administered. One question regarding the use of GO is what level of CD33 expression is necessary to achieve survival benefit. In the frontline COG study, AALL1831, all patients receive GO regardless of CD33 expression, however, it may lack clinical benefit in patients with low expression [82]. Other ADCs currently under investigation are available in adult clinical trials but do not yet extend to the pediatric population.

## 5. Conclusions

Immunotherapy development has increased rapidly in the last decade, and a number of novel and safe and effective therapies are gradually moving to the frontline in pediatric leukemia [83]. The outcome of patients with relapsed B-ALL has considerably improved with CD19 targeting CAR T cells and BiTEs, and the CD22 ADC InO has contributed to this progress. However, other ADC therapies for B-ALL have only shown limited success in comparison, despite promising efficacy in NHL. T-ALL continues to present a challenge for the development of ADC therapy, primarily due to the lack of viable targets. In AML, specific target discovery has been a challenge too, and many tested agents cause unacceptable toxicities. GO experienced a not so straightforward path to FDA approval after concerns of excessive toxicity and limited efficacy, but is now considered a standard component of backbone therapy for de novo AML. Several trials of newer ADCs described above have been abandoned due to a high rate of toxicities and/or lack of clinical response. Despite these setbacks, there continues to be significant interest in ADCs, and refinements of the various components, particularly the linkers and payloads are likely to improve the therapeutic window for these agents. Detailed analyses of the pharmacokinetics and pharmacodynamics of the individual ADCs will guide dosing regimens. The continued search for optimal antigens with greater specificity to the leukemic phenotype compared to normal is leading to preclinical and early clinical studies targeting CD56, CD74, CD276 (B7-H3), CLL1, and FOLR1 among others [84,85,86]. It remains to be seen whether targeting multiple cell surface antigens will improve cytotoxicity and overcome antigen loss. In addition, the optimal combination with standard chemotherapeutic agents and sequence of therapies to maximize efficacy and minimize overlapping toxicities are areas of continued investigation.

## Figures and Tables

**Figure 1 jcm-10-03556-f001:**
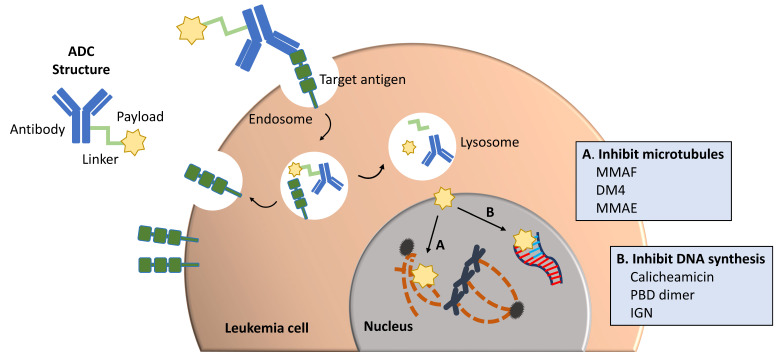
Antibody–drug conjugate structure and mechanisms of action. ADC structure demonstrated on the left consisting of a monoclonal antibody attached to a cytotoxic agent, or payload, by a chemical linker. The ADC binds to the target antigen on the leukemia cell surface, is internalized in an endosome, and then the linker is cleaved from the antibody and payload by chemical or enzymatic reaction in a lysosome. The payload inhibits (**A**) microtubule formation and function or (**B**) DNA synthesis leading to leukemia cell death. The target antigen is recycled back to the cell surface. MMAF, monomethyl auristatin F; DM4, ravtansine; MMAE, monomethyl auristatin E; PBD, pyrrolobenzodiazepine; IGN, indolinobenzodiazepine.

**Table 1 jcm-10-03556-t001:** Active clinical trials utilizing ADCs for acute leukemia.

ADC	Target	Payload	Cancer Targeted	Phase	Age Group	Trial Design	Identifier
Inotuzumab ozogamicin (InO) Besponsa	CD22	Calicheamicin	R/R B-ALL	I	Adult	ALL 001. InO with 3 and 4 drug augmented BFM	NCT03962465
			R/R B-ALL	I	Adult	InO with DA-EPOCH	NCT03991884
			R/R ALL	I/II	Adult	InO post HSCT	NCT03104491
			Upfront ALL	I/II	Adult	InO with low dose chemotherapy	NCT01371630
			R/R B-ALL	I/II	Adult	InO with liposomal vincristine	NCT03851081
			R/R-ALL	I/II	Adult	InO with bosutinib in Ph+ leukemia	NCT02311998
			R/R B-ALL	II	Adult	InO with blinatumomab	NCT03739814
			R/R B-ALL	II	Adult	InO pre and post HSCT	NCT03856216
			R/R B-ALL	II	Adult	InO for MRD positive ALL	NCT03441061
			Upfront ALL	II	Adult	Hyper-CVAD with blinatumomab and inotuzumab	NCT02877303
			Upfront B-ALL	III	Adult	InO with frontline therapy	NCT03150693
			R/R B-ALL	IV	Adult	Varying doses of InO before HSCT	NCT03677596
			Upfront ALL	II	Adult	InO induction followed by conventional chemotherapy	2016-004836-39
			Upfront B-ALL, MPAL, B-LLy	III	Pediatric	AALL1732. InO with standard chemotherapy	NCT03959085
			R/R B ALL	II	Pediatric	InO for MRD positive B ALL	NCT03913559
			Upfront B-ALL	III	Pediatric	ALLTogether1; InO with chemotherapy	NCT04307576
ADCT602 Epratuzumab tesirine	CD22	PBD dimer	R/R ALL	I/II	Adult	Single-agent ADCT602	NCT03698552
Gemtuzumab ozogamicin (GO) Mylotarg	CD33	Calicheamicin	R/R AML	Ib	Adult	BTCRC-AML17-113; GO and venetoclax	NCT04070768
			R/R AML	I	Adult	CPX-351 and GO	NCT03904251
			FLT3 AML	I	Adult	GO, midostaurin, and chemotherapy	NCT03900949
			Upfront core-binding factor AML, MDS	II	Adult	GO with chemotherapy	NCT00801489
			R/R AML	Ib/II	Adult	OX40 antibody alone or in combination with GO or other agents	NCT03390296
			R/R AML	I/II	Adult	Talazoparib with GO	NCT04207190
			Upfront APL	II	10 years and older	Tretinoin and arsenic with or without GO	NCT01409161
			R/R AML	II	Adult	GO with chemotherapy	NCT04050280
			R/R AML	II	Adult	Liposomal daunorubicin, cytarabine, and GO	NCT03672539
			R/R AML, MDS	II	Adult	Single-agent GO for MRD	NCT03737955
			R/R AML	II	Adult	Mitoxantrone, etoposide with GO	NCT03839446
			Upfront AML	III	Adult	GO with chemotherapy, with or without Glasdegib	NCT04093505
			R/R AML	II	Adult	GO with bortezomib and high dose cytarabine	NCT04173585
			Upfront AML	I	Pediatric	GO with standard chemotherapy	NCT04326439
			Upfront AML	III	Pediatric	AAML1831; GO with standard chemotherapy compared to CPX-351 and/or gilteritinib	NCT04293562
			R/R AML	IV	Pediatric	Single-agent GO	NCT03727750
			Upfront AML	III	Pediatric	Myechild01; GO with chemotherapy	NCT02724163
Brentuximab vedotin	CD30	MMAE	Upfront Adult T cell leukemia and lymphoma	II	Adult	Brentuximab with chemotherapy	NCT03264131
IMGN632	CD123	IGN	R/R AML, ALL, BPDCN	I/II	Adult	IMGN632 as monotherapy	NCT03386513
			R/R AML and upfront	Ib/II	Adult	IMGN643 as monotherapy or in combination with venetoclax and/or azacytidine	NCT04086264
VSL-101	ROR1	MMAE	R/R hematologic malignancies	I	Adult	Single-agent VSL-101	NCT03833180

R/R, relapsed refractory; ALL, acute lymphoblastic leukemia; AML, acute myeloid leukemia; BFM, Berlin-Frankfurt-Munster; Ph+, Philadelphia chromosome-positive; HSCT, hematopoietic stem cell transplant; CVAD, cyclophosphamide, vincristine, doxorubicin, and dexamethasone; MRD, minimal residual disease; MPAL, mixed phenotype acute leukemia; B-LLy, B lymphoblastic lymphoma.; APL, acute promyelocytic leukemia; BPDCN, blastic plasmacytoid dendritic cell neoplasm.

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
