# Peer review of "Antibody–Drug Conjugates for the Treatment of Acute Pediatric Leukemia"

_jcm, 2021, doi:10.3390/jcm10163556_

Round 1
Reviewer 1 Report
I suggest discussion of reference Daver #1 in my list in stead of #56 in the manuscript.
Why is moxetumomab pasudotox (ref. #2 #3 in my list not included in the CD22 section?
1: Daver N, Salhotra A, Brandwein JM, Podoltsev NA, Pollyea DA, Jurcic JG,
Assouline S, Yee K, Li M, Pourmohamad T, Samineni D, Sumiyoshi T, Vaze A, Dere
RC, Ma C, Cooper J. A Phase I dose-escalation study of DCLL9718S, an antibody-
drug conjugate targeting C-type lectin-like molecule-1 (CLL-1) in patients with
acute myeloid leukemia. Am J Hematol. 2021 May 1;96(5):E175-E179. doi:
10.1002/ajh.26136. Epub 2021 Mar 11. PMID: 33617672; PMCID: PMC8252033.
2: Shah NN, Schneiderman J, Kuruvilla D, Bhojwani D, Fry TJ, Martin PL, Schultz
KR, Silverman LB, Whitlock JA, Wood B, Vainshtein I, Adams A, Confer D,
Pulsipher MA, Chaudhury S, Wayne AS. Fatal capillary leak syndrome in a child
with acute lymphoblastic leukemia treated with moxetumomab pasudotox for pre-
transplant minimal residual disease reduction. Pediatr Blood Cancer. 2021
Jan;68(1):e28574. doi: 10.1002/pbc.28574. Epub 2020 Sep 22. PMID: 32959985.
3: Shah NN, Bhojwani D, August K, Baruchel A, Bertrand Y, Boklan J, Dalla-Pozza
L, Dennis R, Hijiya N, Locatelli F, Martin PL, Mechinaud F, Moppett J, Rheingold
SR, Schmitt C, Trippett TM, Liang M, Balic K, Li X, Vainshtein I, Yao NS, Pastan
I, Wayne AS. Results from an international phase 2 study of the anti-CD22
immunotoxin moxetumomab pasudotox in relapsed or refractory childhood B-lineage
acute lymphoblastic leukemia. Pediatr Blood Cancer. 2020 May;67(5):e28112. doi:
10.1002/pbc.28112. Epub 2020 Jan 15. PMID: 31944549; PMCID: PMC7485266.
Author Response
Reviewer 1:
- I suggest discussion of reference Daver #1 in my list instead of #56 in the manuscript.
Response: We thank the reviewer for this suggestion. We have added a section on ADC targeting CLL-1 (page 10) and have included the reference Daver et al (Am J Hemat 2021, ref #73). We have kept reference #56 Daver et al Blood ASH Abstract 2018 (now #65) as it adds information regarding an actively recruiting trial for IMGN632, the CD123 targeting ADC.
- Why is moxetumomab pasudotox (ref. #2 #3 in my list not included in the CD22 section?
Response: We thank the reviewer for this suggestion. We originally had a section on immunotoxins but decided to omit this and instead focus on antibody drug conjugates (our assigned topic) as they differ a little in design and mechanism of action. Additionally, the majority of immunotoxins have had little clinical success and excessive toxicities in patients with ALL and AML. However, we have now added moxetumomab (as well as references 2 and 3) into the CD22 targeting section on page 6.
References
1: Daver N, Salhotra A, Brandwein JM, Podoltsev NA, Pollyea DA, Jurcic JG,
Assouline S, Yee K, Li M, Pourmohamad T, Samineni D, Sumiyoshi T, Vaze A, Dere
RC, Ma C, Cooper J. A Phase I dose-escalation study of DCLL9718S, an antibody-
drug conjugate targeting C-type lectin-like molecule-1 (CLL-1) in patients with
acute myeloid leukemia. Am J Hematol. 2021 May 1;96(5):E175-E179. doi:
10.1002/ajh.26136. Epub 2021 Mar 11. PMID: 33617672; PMCID: PMC8252033.
2: Shah NN, Schneiderman J, Kuruvilla D, Bhojwani D, Fry TJ, Martin PL, Schultz
KR, Silverman LB, Whitlock JA, Wood B, Vainshtein I, Adams A, Confer D,
Pulsipher MA, Chaudhury S, Wayne AS. Fatal capillary leak syndrome in a child
with acute lymphoblastic leukemia treated with moxetumomab pasudotox for pre-
transplant minimal residual disease reduction. Pediatr Blood Cancer. 2021
Jan;68(1):e28574. doi: 10.1002/pbc.28574. Epub 2020 Sep 22. PMID: 32959985.
3: Shah NN, Bhojwani D, August K, Baruchel A, Bertrand Y, Boklan J, Dalla-Pozza
L, Dennis R, Hijiya N, Locatelli F, Martin PL, Mechinaud F, Moppett J, Rheingold
SR, Schmitt C, Trippett TM, Liang M, Balic K, Li X, Vainshtein I, Yao NS, Pastan
I, Wayne AS. Results from an international phase 2 study of the anti-CD22
immunotoxin moxetumomab pasudotox in relapsed or refractory childhood B-lineage
acute lymphoblastic leukemia. Pediatr Blood Cancer. 2020 May;67(5):e28112. doi:
10.1002/pbc.28112. Epub 2020 Jan 15. PMID: 31944549; PMCID: PMC7485266.
Reviewer 2 Report
This manuscript is a well-written and structured work, but some considerations could improve the article.
Major points:
- The title is very generic and refers to any type of leukemias, but the content and the authors are related to pediatric leukemias. Therefore, a better title could be 'Antibody drug conjugates for the treatment of pediatric acute leukemia'.
- The possibility of feasibility and convenience of treatment with CAR-T cells after the use of blinatumomab should be discussed.
- The introduction of new concepts regarding the research and use of new ADCs, such as DARTs and others in the treatment of acute leukemias and specifically pediatric acute leukemias, could improve the quality of the article.
- Define the current role of ADCs in the therapeutic algorithms of pediatric acute leukemias, both lymphoid and myeloid lineage.
- In order to improve the manuscript I suggest reading the next papers to introduce some addittional information:
- Stephen P. Hunger, Elizabeth A. Raetz. How I treat relapsed acute lymphoblastic leukemia in the pediatric population 2020;136(16):1803-1812).
- Le Li, Ying Wang. Recent updates for antibody therapy for acute lymphoblastic leukemia. Exp Hematol Oncol (2020) 9:33.
- Hiroto Inaba, Charles G. Mullighan. Pediatric acute lymphoblastic leukemia. Haematologica 2020 Volume 105(11):2524-2539.
- Jing Chen, Chana L. Glasser. New and Emerging Targeted Therapies for Pediatric Acute Myeloid Leukemia (AML). Children 2020, 7, 12
Minor points:
To correct
- Inotuzumab (line 14)
- To include Pediatric (pediatric acute leukemia), at the end of abstract (line 21)
- Daratumumab (line 214)
- Figure 1. To define MMAF, DM4, MMAE, PBD dimers, ING.
Author Response
Reviewer 2:
This manuscript is a well-written and structured work, but some considerations could improve the article.
Major points:
- The title is very generic and refers to any type of leukemias, but the content and the authors are related to pediatric leukemias. Therefore, a better title could be 'Antibody drug conjugates for the treatment of pediatric acute leukemia'.
Response: We thank the reviewer for this suggestion, we have now included the word “pediatric” in the title.
- The possibility of feasibility and convenience of treatment with CAR-T cells after the use of blinatumomab should be discussed.
Response: We do not discuss BITEs or CAR T cells in this article as our assigned topic for this review is antibody drug conjugates. Another article in this series on pediatric leukemia will focus specifically on BITEs and CAR T cells.
- The introduction of new concepts regarding the research and use of new ADCs, such as DARTs and others in the treatment of acute leukemias and specifically pediatric acute leukemias, could improve the quality of the article.
Response: DARTs and other antibody engagers such as BITEs will also be discussed in a separate article in this series of articles on pediatric leukemia.
- Define the current role of ADCs in the therapeutic algorithms of pediatric acute leukemias, both lymphoid and myeloid lineage.
Response: We thank the reviewer for this suggestion. We have added a paragraph at the end of the ALL and AML sections reflecting current clinical applications of ADCs with the help of the articles cited in #5.
- In order to improve the manuscript I suggest reading the next papers to introduce some addittional information:
- Stephen P. Hunger, Elizabeth A. Raetz. How I treat relapsed acute lymphoblastic leukemia in the pediatric population 2020;136(16):1803-1812).
- Le Li, Ying Wang. Recent updates for antibody therapy for acute lymphoblastic leukemia. Exp Hematol Oncol (2020) 9:33.
- Hiroto Inaba, Charles G. Mullighan. Pediatric acute lymphoblastic leukemia. Haematologica 2020 Volume 105(11):2524-2539.
- Jing Chen, Chana L. Glasser. New and Emerging Targeted Therapies for Pediatric Acute Myeloid Leukemia (AML). Children 2020, 7, 12.
Response: We have added Hunger et al, Blood (ref#37) and Inaba et al Haematologica (ref#39) to assist with suggestion #4 and we had previously included Li et al, Exp Hematol Oncol (ref#20) and Chen et al, (ref#70).
- Minor points:
To correct
- Inotuzumab (line 14)
- To include Pediatric (pediatric acute leukemia), at the end of abstract (line 21)
- Daratumumab (line 214)
- Figure 1. To define MMAF, DM4, MMAE, PBD dimers, ING.
Response: We thank the reviewer for noticing these errors and omissions. We have updated and addressed each one in the manuscript.
Reviewer 3 Report
Dear Authors
The article is very interesting. The review focuses on using ADCs for the treatment of acute leukemia. The Authors should emphasize more the role of ADCs in the treatment of leukemias in children. For the reader is not clear what is the position of ADCs in pediatric oncology, what are the detailed indications for using ADCs in treatment. The second problem are side affects of ADCs.
Could the Authors extend this information.
Author Response
Dear Authors
The article is very interesting. The review focuses on using ADCs for the treatment of acute leukemia. The Authors should emphasize more the role of ADCs in the treatment of leukemias in children. For the reader is not clear what is the position of ADCs in pediatric oncology, what are the detailed indications for using ADCs in treatment. The second problem are side affects of ADCs.
Could the Authors extend this information.
Response: We thank the reviewer for the feedback. We have added a section on the current clinical applications of ADCs in pediatric ALL and AML at the end of each respective section. We have expanded the general mechanisms of toxicities in section 1.2. Specific toxicities of each ADC are described in the respective sections.
Round 2
Reviewer 2 Report
After review, I suggest the acceptance of the manuscript. Congrats!